# Sodium-Glucose Cotransporter-2 Inhibitors in Diabetes and Beyond: Mechanisms, Pleiotropic Benefits, and Clinical Use—Reviewing Protective Effects Exceeding Glycemic Control

**DOI:** 10.3390/molecules30204125

**Published:** 2025-10-18

**Authors:** Julia Hanke, Katarzyna Romejko, Stanisław Niemczyk

**Affiliations:** Department of Internal Diseases, Nephrology and Dialysis, Military Institute of Medicine—National Research Institute, 128 Szaserów Street, 04-141 Warsaw, Poland; kromejko@wim.mil.pl (K.R.); sniemczyk@wim.mil.pl (S.N.)

**Keywords:** SGLT2 inhibitors, glycemic control, renoprotective role, heart failure, hepatic effects, cognitive function, metabolic regulation

## Abstract

Sodium-glucose cotransporter-2 (SGLT2) inhibitors, also known as gliflozins, are a class of antidiabetic agents that act independently of insulin by promoting renal glucose excretion. They modulate glucose reabsorption in proximal renal tubules. Initially, they were used for the treatment of type 2 diabetes mellitus (T2DM); however, numerous pleiotropic benefits beyond glycemic control were observed. Large clinical trials confirmed their efficacy in reducing cardiovascular mortality, heart failure hospitalizations, and progression of chronic kidney disease. SGLT2 inhibitors reduce oxidative stress and inflammation and induce favorable metabolic adaptations, including lowering ketosis and upregulation of erythropoiesis. They also exert protective effects on hepatic and cognitive function. Additionally, SGLT2 inhibitors lower serum uric acid and reduce adipose tissue mass, which usually results in weight loss. Although generally well-tolerated, they are associated with increased risk of urogenital infections, euglycemic ketoacidosis, and a potentially enlarged amputation risk. Current guidelines worldwide recommend their use not only for T2DM but also for heart failure and chronic kidney disease, marking a paradigm shift toward organ-protective therapies. This review provides a comprehensive synthesis of current evidence on the mechanisms, clinical benefits, and safety profile of SGLT2 inhibitors, highlighting their expanding role in cardiometabolic and multisystem disease management.

## 1. Introduction

Sodium-glucose cotransporter-2 (SGLT2) inhibitors, also known as gliflozins, are a class of antidiabetic drugs initially developed to improve glycemic control in patients with type 2 diabetes mellitus (T2DM) [1,2,3]. They promote the excretion of glucose excess through the kidneys, resulting in significant glucosuria and effectively lowering plasma glucose levels independently of insulin secretion or action [4,5]. In addition to their primary role in glycemic control, SGLT2 inhibitors have several other beneficial effects, including weight loss, reduction of cardiovascular risk, and kidney protection [6,7,8,9,10,11,12,13]. Recent large-scale clinical trials showed that empagliflozin, dapagliflozin, and canagliflozin significantly improve cardiovascular and renal outcomes, leading to recommendations for their use in high-risk populations, such as those with heart failure and chronic kidney disease [9,11,14,15,16]. Ongoing research continues to explore the broader therapeutic potential of SGLT2 inhibitors, including their applications in metabolic syndrome, liver diseases, and even neurodegenerative disorders [17,18,19,20,21,22,23]. Given their significant clinical implications and wide-ranging benefits, SGLT2 inhibitors represent a major advancement in medicine, highlighting the need for a thorough understanding of their complex mechanisms and diverse therapeutic uses. Given the substantial, pleiotropic potential of these drugs, we opted to conduct a scope review to consolidate all the advantageous features and elucidate the mechanisms of action of these promising agents [24,25,26].

## 2. Historical Background and Development of SGLT2 Inhibitors

Phlorizin (glucose, 1-[2-(β-D-glucopyranosyloxy)-4,6-dihydroxyphenyl]-3-(4-hydroxyphenyl)-1-propanone) was the first substance identified as a member of the gliflozin class. It was isolated in 1835 by a French chemist from the bark of the apple tree (Malus domestica). Initially, due to its bitter taste, phlorizin was presumed to possess antipyretic properties and was explored as a potential treatment for malaria and various infectious diseases [27]. The glucose-lowering effect of phlorizin, manifested as glucosuria, was first documented by von Mering in 1886 [28].

As the investigation progressed, the mechanisms of glucose reabsorption in the glomerulus were elucidated, demonstrating that active transport is essential for this process. It became evident that glucose is cotransported with sodium. It became apparent that the inhibition of this active co-transporter results in the excretion of both sodium and glucose in the urine [24,25].

During the 1960s, the non-selective inhibitor of both sodium-glucose cotransporter-1 (SGLT1) and SGLT2, known as phlorizin, drew interest as a possible treatment for diabetes [24,25]. Preclinical studies conducted in the 1980s demonstrated that phlorizin enhanced insulin sensitivity, increased urinary glucose excretion, and normalized plasma glucose concentrations in diabetic rats, without causing hypoglycemia [29]. Despite these promising effects, phlorizin was unsuitable for clinical use due to poor oral bioavailability and non-selectivity for SGLT2 [1].

The first orally bioavailable, chemically engineered phlorizin analog was T-1095, developed in the 1990s. It demonstrated some efficacy in lowering glycated hemoglobin (HbA1c) levels, reducing microalbuminuria, and promoting weight loss in rats. However, due to its lack of selectivity for SGLT2 and adverse gastrointestinal effects, T-1095 did not advance to clinical use [30].

A significant breakthrough occurred in 2012 when dapagliflozin was approved by the European Medicines Agency (EMA) as the first selective SGLT2 inhibitor for the treatment of T2DM [31].

This marked the beginning of a new class of antidiabetic agents offering insulin-independent glycemic control. In 2013, the United States Food and Drug Administration (FDA) granted approval for canagliflozin, followed by subsequent authorizations for other agents within the class, including empagliflozin, ertugliflozin, dapagliflozin, ipragliflozin, luseogliflozin, tofogliflozin, remogliflozin, and many others [14,31,32,33].

## 3. Mechanism of Action and Pharmacodynamics of SGLT2 Inhibitors

### 3.1. SGLT’s Function in Renal Glucose Homeostasis

Sodium-glucose transporters (SGLTs) are integral membrane proteins responsible for maintaining glucose homeostasis by actively reabsorbing glucose from the primary filtrate in the renal proximal tubule. Among the SGLT family, two isoforms—SGLT1 and SGLT2—are of particular physiological importance. SGLT2 is predominantly expressed in the luminal membrane of the S1 and S2 segments of the proximal convoluted tubule (PCT), where it reabsorbs the majority of filtered glucose. In contrast, SGLT1 is more widely distributed, with its prominent expression in the small intestine, heart, and skeletal muscle, and contributes to glucose reabsorption in the more distal S3 segment of the PCT [34].

Functionally, SGLT1 mediates sodium-dependent transport of both glucose and galactose, whereas SGLT2, despite exhibiting a similar transport mechanism, has a different preference for substrates—it exclusively transports glucose and has a low affinity for galactose [34,35]. In healthy individuals, the coordinated action of SGLT2 and SGLT1 enables the reabsorption of more than 99% of the filtered glucose load, which is approximately 180 g per day, effectively preventing glucose loss in the urine [34,36]. SGLT2 accounts for approximately 90–97% of total glucose reabsorption. Although it exhibits low glucose affinity, it has high transport capacity. The remaining glucose load is managed by SGLT1, which compensates with its higher affinity for glucose [2].

This synergistic mechanism allows dynamic adaptation to daily fluctuations in plasma glucose concentration. Renal glucose reabsorption is directly proportional to the filtered glucose load, up to a maximum threshold known as TmaxG, typically reached at plasma glucose concentrations of 180–200 mg/dL [37]. In healthy adults, the maximal glucose reabsorptive capacity (TmG) is estimated at approximately 375 mg/min, although this value can vary across populations and depending on methodological factors [4]. The renal threshold for glucose, defined as the plasma glucose concentration at which SGLT-mediated transport becomes saturated, can increase significantly in patients with poorly controlled diabetes. In some cases, the threshold has been reported to rise to as high as 700 mg/dL, contributing to enhanced total glucose reabsorption of up to 500–600 g daily [4,38,39].

This adaptive response is mediated by the upregulation of SGLT2 expression and increased activity of the basolateral Na^+^/K^+^-ATPase pump, which maintains the necessary sodium gradient. However, such compensation also imposes an increased metabolic demand on tubular cells, potentially leading to cellular stress and injury, which may contribute to the development of diabetic kidney disease [40]. In the kinetic model, SGLTs are characterized as double-gated transmembrane proteins [34,41]. The sodium concentration gradient—high in the tubular lumen and low in the cytoplasm—provides the driving force for glucose transport against its concentration gradient. Sodium ions first bind to the extracellular domain of the transporter, initiating a conformational change that increases the affinity for glucose. Once glucose binds, another conformational shift occurs, allowing sodium to move down its electrochemical gradient into the cytoplasm. As it happens, the transporter’s affinity for glucose decreases, and it is subsequently released into the intracellular space [42]. The sodium concentration gradient is maintained by the basolateral Na^+^/K^+^-ATPase, which extrudes three sodium ions into the cell in exchange for two potassium ions, demonstrating secondary active transport [2,3,25,37]. Remarkably, SGLT1 operates with a stoichiometry of 2 Na^+^ to 1 glucose molecule, while SGLT2 functions with a 1:1 ratio. This difference reflects their distinct physiological roles in tissues and their adaptation to substrate availability [25]. The final phase in glucose reabsorption involves its passive diffusion through the basolateral glucose transporter 2 (GLUT2) transporter, down the concentration gradient, into the interstitial fluid and bloodstream (Figure 1) [2,3,25,37].

### 3.2. Renal Effects of SGLT2 Blockade

Currently available drugs used in clinical practice from the gliflozin class are selective, competitive inhibitors of the SGLT2 [43]. Their primary mechanism relies on blocking glucose reabsorption in PCT, thus promoting urinary glucose excretion and regulating the glycemic serum concentration. Therapeutic agents designed to target this transporter must exhibit a high degree of selectivity for SGLT2 over SGLT1, as inhibition of SGLT1 alone can lead to severe diarrhea and dehydration. This is essential to preserve patients’ gastrointestinal integrity and safety [25]. Although sotagliflozin, as a dual SGLT1 and SGLT2 inhibitor, has been administered as an addition to insulin therapy in patients with type 1 diabetes, improvements not only in glycemic control but also in body weight and blood pressure regulation were described. These findings indicate a potential therapeutic benefit in selected cases [44]. Under complete SGLT2 inhibition, glucose reabsorption in the S1 and S2 segments of PCT is effectively abolished. As a result, the entire filtered glucose load flows to the S3 segment, where SGLT1 transporters are located [4]. Current literature suggests that under physiological settings, SGLT1 functions well below its maximum transport capacity, reabsorbing only 10–15% of the filtered glucose. However, when SGLT2 is entirely inhibited to mitigate excessive energy loss with glucosuria, SGLT1 is forced to function at its maximal transport capacity [45]. This means that SGLT1 can reabsorb up to approximately 120 g of glucose per day. Only the portion of filtered glucose that exceeds this threshold is excreted in the urine. Consequently, even at maximal doses of SGLT2 inhibitors, which fully inhibit the main glucose transporter, only 50–60 g of glucose per day is observed to be excreted in the urine in people without carbohydrate metabolism disorders. This mechanism explains why increasing the dose of an SGLT2 inhibitor tenfold does not proportionally increase glucosuria [4,45].

### 3.3. Pharmacodynamic Outcomes and Glycemic Efficacy

According to a meta-analysis conducted by Monami, SGLT2 inhibitors reduced HbA1c concentrations in T2DM patients by roughly 0.5–0.7% after 12 weeks of treatment. The effect lasted for up to 52 weeks. The overall amount of urine glucose excretion—and consequently the glycemic effectiveness of SGLT2 inhibitors—is greater in people with higher baseline blood glucose levels and intact estimated glomerular filtration rates (eGFR). As a result, the therapeutic efficacy of SGLT2 inhibitors is essentially dependent on kidney function [46].

## 4. Genomic Localization and Structure of SGLT Family Members

For context, the SGLT1 gene (*SLC5A1*) was initially mapped to the q11.2–qter region of chromosome 22 and subsequently refined by fluorescence in situ hybridization (FISH) to the proximal half of band 22q13.1 [47,48]. In contrast, SGLT2 (*SLC5A2*) is located on chromosome 16 (16p12–p11), near the centromeric region [34]. The SGLT2 gene is a single-copy locus with a compact genomic organization (reports indicate ~14 exons), and—similar to other SGLT family members—its overall exon–intron structure is conserved despite substantial variation in total gene length across the family [34,49]. At the structural level, SGLT2 is expected to adopt the canonical SGLT topology, with ~14 transmembrane helices and both the amino (NH_2_) and carboxyl (COOH) termini facing the extracellular side, by homology to SGLT1 [49]. The completion of the Human Genome Project enabled precise assignment of SGLT loci to chromosomes, including SGLT1 (22q13.1), SGLT2 (16p12–p11), SGLT3 (21q22.12), SGLT4 (1p32), SGLT5 (17p11.2), and SGLT6 (16p12.1) [34].

## 5. Effects of SGLT2 Inhibitors—Evidence and Clinical Recommendations

SGLT2 inhibitors exhibit multiple additive mechanisms of action, primarily exerting a positive effect on the cardiovascular system and providing renoprotection. However, their positive impact on the organism does not end there, which will be discussed in this article. The mechanism of their cardioprotective and renoprotective effects is schematically presented in Figure 2.

### 5.1. Improved Glycemic Control

The original therapeutic target of SGLT2 inhibitors was to deal with hyperglycemia in T2DM patients. Improved glycemic control is often represented by a decrease in HbA1c concentrations of approximately 0.5–1% [14,50]. According to Zaccardi’s meta-analysis regarding the three most commonly used SGLT2 inhibitors—canagliflozin, dapagliflozin, and empagliflozin—canagliflozin at a daily dose of 300 mg resulted in the highest reduction in HbA1c. Compared to other SGLT2 inhibitors, the HbA1c-lowering effect of canagliflozin 300 mg exceeded that of dapagliflozin 5 mg by 0.3% (3.3 mmol/mol) and that of empagliflozin 100 mg by 0.1% (1.1 mmol/mol). At different dosages, dapagliflozin and empagliflozin did not differ significantly. In comparison to a placebo, the meta-analysis also verified a significant decrease in fasting plasma glucose (FPG). FPG decreased significantly with canagliflozin 300 mg (−2.0 mmol/L) and least with dapagliflozin 5 mg (−1.1 mmol/L). When compared to other glucose-lowering medications, the differences ranged from −1.2 mmol/L (canagliflozin 300 mg vs. dipeptidyl) peptidase 4 (DPP-4) inhibitors to −0.3 mmol/L (empagliflozin 10 mg vs. metformin). These results strengthen the concept that canagliflozin 300 mg is the best SGLT2 inhibitor for reducing FPG [50,51].

SGLT2 inhibitors, when used in conjunction with insulin therapy, contribute to more stable 24-h glycemic profiles. The proportion of time spent in the normoglycemic range (70–180 mg/dL) increases, without elevating the risk of hypoglycemia (<70 mg/dL). Simultaneously, the frequency of hyperglycemia (>180 mg/dL) decreases, resulting in more constant metabolic management, which not only decreases the glucose toxicity but also promotes weight loss and enables reducing insulin doses [52,53]. Apart from their antihyperglycemic characteristics, SGLT2 inhibitors also improve pancreatic β-cell function, increase insulin sensitivity in peripheral tissues, and moderately reduce endogenous insulin synthesis. These may delay the need to implement insulin therapy in T2DM patients [5,54,55,56]. The results of significant clinical trials on SGLT2 inhibitors support their ability to achieve glycemic control when used alone or alongside current anti-diabetic therapy [8,9,11,52,53,57].

Studies that used SGLT2 inhibitors in addition to insulin regimens showed that the amount of time spent with normoglycemia (glucose levels between 70 and 180 mg/dL) was significantly higher with luseogliflozin than with a placebo [53]. There was also a reported decrease in the number of insulin units required to maintain a normal blood glucose level [52].

### 5.2. Weight Loss and Lipid Metabolism

For patients with T2DM and concomitant obesity, according to the American Diabetes Association (ADA), SGLT2 inhibitors are recommended as one of the first-line pharmacologic options [58]. Moderate weight reduction has been demonstrated while using these agents, typically in the range of 2 to 4 kg over a period of 6 to 12 months [26,59]. Notably, during the first week of treatment, patients may have a rapid weight loss of up to 1 kg, due to osmotic diuresis and fluid loss caused by glucosuria [57]. Beyond this early fluid-mediated effect, sustained weight loss is suspected to be driven by increased urinary glucose excretion and a negative caloric balance. A daily glucosuria of 60–80 g of glucose leads to a loss of an estimated 1000–1300 kJ (230–310 kcal) per day, regardless of additional physical activity or dietary restrictions. Mathematical modeling predicts that in the absence of caloric compensation, the negative energy balance induced by SGLT2 inhibition could lead to a weight loss of approximately 7 kg over six months. However, real-world data consistently demonstrate that early weight reduction is followed by stabilization, even with ongoing therapy [51,57].

At the cellular level, SGLT2 inhibition enhances renal gluconeogenesis in the proximal tubule. By disrupting the physiological process of glucose detection and creating a disparity between primary urine glucose levels and reabsorbed glucose levels, SGLT inhibition raises the expression of the gluconeogenesis gene [60].

Hypersecretion of glucagon from pancreatic α-cells is a compensatory reaction for SGLT2 inhibition. This response is mediated by decreased intracellular glucose concentrations and subsequent activation of K-ATP channels. What is interesting, both SGLT1 and SGLT2 transporters are expressed in α-cells, which are responsible for transporting glucose from serum into the cells, but are absent in β-cells. This supports the selectivity of this mechanism. Experimental models confirm that SGLT2 blockade elevates glucagon expression via a feedback loop involving the glucagon receptor pathway. In animal models, SGLT2 inhibition has been shown to increase circulating glucagon levels. Elevated glucose concentrations downregulate *SLC5A2*—gene coding SGLT2—transcription and upregulate glucagon gene (GCG) transcription. This leads to increased GCG mRNA levels in human pancreatic islets. These results are consistent with previous studies indicating that glucagon release from α-cells activates a positive feedback loop mediated by glucagon receptor signaling, which further amplifies GCG gene expression [61]. Increased glucagon levels, along with a diminished insulin-to-glucagon ratio, promote hepatic glucose production by enhancing glycogenolysis and consequently gluconeogenesis [57]. Simultaneously, hepatic glycogen depletion and relative hypoinsulinemia favor a metabolic shift toward lipolysis and ketogenesis [5,62]. This adaptive response promotes mobilization and oxidation of free fatty acids from adipose tissue, thereby providing ketone bodies as an efficient alternative energy source [5,57]. Mild nutritional ketosis induced by SGLT2 inhibitors is suggested to enhance cardiovascular health [63]. Moreover, SGLT2 inhibitors may decrease renal ketone clearance by reducing glomerular filtration and enhancing their tubular reabsorption, thus preserving them as a metabolic fuel [64].

Studies in both humans and animal models have demonstrated that SGLT2 inhibitor therapy leads to reductions in both visceral and subcutaneous fat depots [6,65]. Since lean body mass is generally preserved during treatment, it was suggested that the primary cause of weight loss is the reduction in adipose tissue [57,63]. Another suggested mechanism that restricts long-term weight loss is compensatory hyperphagia [7,63]. In T2DM patients, SGLT2 inhibitors have been shown to lower leptin levels. Leptin is a satiety hormone produced by adipocytes; its decline demonstrates fat loss. However, it also stimulates hunger. However, under conditions of enhanced metabolism, this effect does not outweigh treatment advantages [66].

### 5.3. Renoprotective Effects

According to the EMPA-REG OUTCOME trial, empagliflozin lowered the risk of developing or exacerbating nephropathy, defined as progression of macroalbuminuria, doubling of serum creatinine levels by 44%, initiation of renal replacement therapy by 55%, or death from kidney disease by 39%. These effects were still observed despite that over 80% of patients were already receiving angiotensin-converting enzyme inhibitors (ACEI) or angiotensin receptor blockers as part of standard care [11].

SGLT2 inhibitors have a complex renoprotective mechanism that is hard to assign to a single action. One major pathway involves the attenuation of maladaptive sodium and glucose reabsorption in the proximal tubule. This restores tubuloglomerular feedback signaling via the macula densa and reduces hydrostatic pressure in Bowman’s space, ultimately mitigating glomerular hyperfiltration [63]. In early diabetic nephropathy, mild hyperglycemia increases SGLT-mediated sodium and glucose reabsorption, which decreases the amount of sodium delivery to the farther macula densa and triggers afferent arteriolar dilation. The resulting rise in intraglomerular pressure leads to glomerular injury and albuminuria, characteristics of diabetic nephropathy. This maladaptive feedback loop promotes proximal tubular hypertrophy, increases energy demands via Na^+^/K^+^-ATPase activity, and contributes to mitochondrial stress and oxidative damage due to excess intracellular glucose shunted into non-glycolytic pathways [40]. SGLT2 inhibition reverses this loop by increasing distal sodium delivery, constricting the afferent arteriole, and thereby reducing glomerular filtration pressure. The initial decrease in eGFR observed upon therapy initiation reflects this hemodynamic adjustment and typically remains within a clinically acceptable range (2–4 mL/min), with full reversibility upon drug discontinuation [63]. Inhibition of tubuloglomerular feedback, glomerular hyperfiltration, and excessive glucose reabsorption collectively reduces renal oxygen consumption and cellular workload, while improving glucose handling and limiting glucotoxicity. These processes confer long-term protection to renal structure and function [2,17,67].

Emerging evidence suggests that SGLT2 inhibitors, through mechanisms similar to ACEIs, have renoprotective benefits that extend beyond diabetic nephropathy to protein excess. This was depicted in preclinical mouse models, where dapagliflozin ameliorated podocyte injury and cytoskeletal remodeling, mimicking the nephroprotective mechanisms of ACEIs [68].

These two substances share the same hemodynamic effect on the glomerulus. Both reduce intraglomerular pressure and hyperfiltration. As previously stated, SGLT2 inhibitors accomplish this action predominantly by afferent arteriole constriction, whereas ACEI primarily do so by dilatation of the efferent arteriole. An early, mild drop in eGFR, followed by a slower loss of filtration and decreased proteinuria, is a clinically comparable effect [68,69].

Both classes limit podocyte damage: ACEI by blocking harmful angiotensin II signals in the podocyte, and SGLT2 inhibitors by reducing hemodynamic stress. In a protein overload model, dapagliflozin protected podocytes to a degree comparable to lisinopril. Both classes reduce proteinuria, a marker of improved podocyte function [69].

About two-thirds of all sodium reabsorption occurs in the PCT by exchange with hydrogen ions via the sodium/hydrogen exchanger isoform 3 (NHE3). SGLT2 is colocalized with NHE3 in the proximal tubule. Their functions appear to be interdependent, such that SGLT2 inhibition may also downregulate NHE3 activity. Therefore, NHE3 may also be indirectly inhibited by SGLT2 inhibitors. This dual action promotes natriuresis, blood pressure reduction, and improves glomerular hemodynamics [70,71]. Studies in rodents with diabetic kidney disease showed a 40% increase in basal renal oxygen consumption, attributed to tubular hypertrophy, gluconeogenesis, and compensatory increase in the number of SGLT and NHE3 transporters. By increasing osmotic diuresis and decreasing transcellular sodium reabsorption, SGLT2 blockage lessens this oxygen burden [72,73]. Animal studies revealed that SGLT2 inhibition may prevent kidney growth, inflammation, and fibrosis in diabetes through glucose-lowering independent mechanisms. These findings collectively support the view that SGLT2 inhibitors offer substantial renal protection through both hemodynamic and cellular pathways [63,74,75,76,77,78,79,80,81,82,83,84,85,86,87,88,89].

### 5.4. Reduction in Hospitalization and Mortality in Heart Failure

The EMPA-REG OUTCOME trial evaluated the cardiovascular effects of empagliflozin (10 mg or 25 mg once daily) versus placebo in 7020 patients with type 2 diabetes, followed over a median duration of 3.1 years. Compared to placebo, empagliflozin significantly reduced the risk of cardiovascular death by 38% (HR 0.62; 95% CI: 0.49–0.77; *p* < 0.001), all-cause mortality by 32% (HR 0.68; 95% CI: 0.57–0.82; *p* < 0.001), and hospitalization due to heart failure by 35% (HR 0.65; 95% CI: 0.50–0.85; *p* = 0.002). No significant differences were observed in the incidence of myocardial infarction (MI) or stroke [9].

The CANVAS Program, comprising two large-scale trials with 10,142 participants, demonstrated that canagliflozin reduced the risk of the composite cardiovascular endpoint (cardiovascular death, nonfatal MI, or stroke) by 14% (HR 0.86; 95% CI: 0.75–0.97; *p* < 0.001 for non-inferiority; *p* = 0.02 for superiority). However, no statistically significant reduction was found in all-cause mortality (HR 0.87; 95% CI: 0.74–1.01) or cardiovascular death (HR 0.87; 95% CI: 0.72–1.06) [14].

In the DECLARE-TIMI 58 trial, which included 17,160 patients followed for a median of 4.2 years, dapagliflozin significantly reduced the combined risk of cardiovascular death or hospitalization for heart failure by 17% (HR 0.83; 95% CI: 0.73–0.95; *p* = 0.005). This benefit was primarily driven by a 27% reduction in hospitalization for heart failure (HR 0.73; 95% CI: 0.61–0.88). Dapagliflozin did not significantly affect the incidence of major adverse cardiovascular events (MACE), such as cardiovascular death, non-fatal myocardial infarction, and non-fatal stroke. MACE occurred in 8.8% of patients in the dapagliflozin group versus 9.4% in the placebo group (HR 0.93; 95% CI: 0.84–1.03; *p* = 0.17) [10].

SGLT2 inhibitors generate a net negative balance of sodium and water by promoting natriuresis and glucosuria [80]. This reduction in plasma volume is associated with a decrease in systolic blood pressure by approximately 3–6 mmHg and in diastolic pressure by 1–1.5 mmHg [12,14]. Furthermore, these agents may improve vascular function by reducing arterial stiffness, as reflected by lower pulse wave velocity (PWV) [81,82]. Arterial stiffness, a crucial marker of vascular change, is defined as an increase in PWV along the arteries of both forward and backward (reflected) pulse waves, resulting in higher central systolic blood pressure and elevated central pulse pressure [83]. However, this has not been observed in all conducted studies. Other studies are needed to assess the variability of factors on which favorable changes depend [84].

On a metabolic level, SGLT2 inhibitors stimulate the production of ketone bodies, particularly β-hydroxybutyrate (βOHB). βOHB can be an efficient energy substrate for myocardium, especially under the influence of stress or hypoxia [85,86]. βOHB improves mitochondrial function and makes ATP more efficient. Compared to glucose and fatty acids, βOHB delivers more ATP per molecule of oxygen and generates fewer reactive oxygen species (ROS), reducing oxidative damage to mitochondrial DNA and respiratory chain proteins. Moreover, βOHB is a signaling molecule. By inhibiting histone deacetylases (HDACs), it promotes the expression of genes responsible for energy production and antioxidant defense [87,88]. Ketone bodies may reduce sympathetic nervous system activity, resulting in lower heart rate and blood pressure [89]. Another advantage conferred by βOHB is the activation of AMP-activated protein kinase (AMPK), which reestablishes energy homeostasis by blocking the mammalian target of the rapamycin (mTOR) signaling pathway, a recognized promoter of heart hypertrophy and fibrosis. Activation of AMPK maintains mitochondrial membrane potential, suppresses apoptosis through the B cell Lymphoma 2 (Bcl-2) pathway, and enhances cardiomyocyte viability [87].

SGLT2 inhibitors also activate the transcription factor nuclear factor erythroid 2-related factor 2 (Nrf2), which increases the expression of key antioxidant enzymes such as include superoxide dismutase (SOD), peroxiredoxins (PRDX), glutathione peroxidase (GPX), and heme oxygenase-1 (HO-1), which further contributes to cardiomyocyte protection [90].

Additionally, SGLT2 inhibitors increase nitric oxide (NO) bioavailability, thereby improving endothelial function and enhancing coronary blood flow [85].

Emerging evidence suggests a lower risk of arrhythmias during SGLT2 inhibitor therapy [91]. Several pathways have been proposed to explain this effect. As excessive myocardial strain and hypertrophy have been associated with an increased incidence of arrhythmias, one study proposes that reducing myocardial stretch through decreased plasma volume may minimize cardiac arrhythmogenesis [91,92]. Additionally, by reducing oxidative stress and ROS production, preserving mitochondrial integrity, and elevating βOHB levels, SGLT2 inhibitors can stabilize the cell/membrane potential and exert an antiarrhythmic effect [63,64]. The disturbance of Na^+^ and Ca^2+^ homeostasis that occurs in heart failure (HF) contributes to arrhythmogenesis. Thus, SGLT2 inhibitors restore ion balance and lower the incidence of arrhythmias [93,94].

The effects of SGLT2 inhibitors on the lipid profile still remain complex and inconsistent. Some data indicate that cardiovascular advantages arise from a decreased atherogenic lipid profile [95]. Extensive meta-analyses have indicated a modest increase in total cholesterol levels. In two pooled analyses involving 24,782 and 147,130 participants, the average increase in total cholesterol was 0.09 mmol/L [96,97]. However, some observational or retrospective studies reported no significant changes, and a few even indicated small reductions [98,99]. The impact of SGLT2 inhibitors on low-density lipoprotein (LDL) cholesterol is equally complex. Two meta-analyses showed a small but statistically significant increase in LDL cholesterol. In one study that included 48 trials, the rise was 0.10 mmol/L (95% CI: 0.07–0.12), and 0.08 mmol/L (2.92 mg/dL) in the larger meta-analysis by Yaribeygi [90,96]. In contrast, some trials, including post-hoc analyses using empagliflozin, reported no major changes in LDL cholesterol levels, while others reported a decrease [95,98,100]. However, qualitative analysis of LDL particle composition changes may underlie cardiovascular benefits. SGLT2 inhibitors may reduce tiny, dense LDL particles, which are more atherogenic, while increasing bigger particles, despite an increase in overall LDL cholesterol [95]. According to a review by Osto, this shift offsets any negative effect of increased LDL levels and supports the overall cardioprotective profile of these agents [101]. SGLT2 inhibitors have consistently demonstrated a tendency to increase high-density lipoprotein (HDL) cholesterol concentrations [96,97]. For triglycerides, the effect of SGLT2 inhibitors is more consistent. A meta-analysis of 48 trials (−0.10 mmol/L; 95% CI: −0.13 to −0.07) and another involving 60 studies (−0.10 mmol/L or −8.78 mg/dL) demonstrated that SGLT2 inhibitors effectively reduce triglyceride concentrations [88,90]. Interestingly, this triglyceride-lowering impact appears to be stronger in Asian populations [97]. Basu, using a transgenic mouse model, demonstrated that SGLT2 inhibition enhances lipoprotein lipase (LPL) activity, leading to accelerated hydrolysis of triglyceride-rich lipoproteins, including very low-density lipoprotein (VLDL) and chylomicrons. Enhanced LPL activity increases free fatty acid release and promotes more rapid VLDL clearance [102]. Since VLDL is a precursor to LDL, increased VLDL metabolism was associated with higher LDL particle formation. LDL particles also seemed to be removed from the bloodstream less quickly [102]. This could be related to diminished expression of LDL receptors in the liver as a result of decreased insulin signaling. The reduced insulin levels during SGLT2 inhibitor therapy could potentially moderately increase the levels of LDL cholesterol in the circulation, as insulin encourages hepatocytes to absorb LDL [103]. Nonetheless, the overall lipoprotein profile appears to shift in a positive direction. The presence of larger, less atherogenic LDL particles, as well as reductions in triglycerides and increases in HDL cholesterol, supports the hypothesis that SGLT2 inhibitors have a cardiometabolically neutral or even favorable effect on lipid metabolism [101,104].

Key results and safety signals from the largest randomized trials are summarized in Table 1.

### 5.5. Inhibition of Hyperuricemia

Uric acid (UA) has been increasingly recognized as a pro-inflammatory mediator enhancing oxidative stress and activating the renin–angiotensin–aldosterone system (RAAS) [116,117].

From a clinical perspective, elevated serum UA levels are closely associated with a spectrum of metabolic abnormalities—including insulin resistance and hyperglycemia—as well as cardiovascular conditions such as hypertension, endothelial dysfunction, arterial stiffness, and HFpEF. Additionally, diabetic nephropathy, one of the most common consequences of T2DM, is associated with hyperuricemia both in its origin and progression [116,117,118].

Approximately 90% of UA filtered through the glomerulus is reabsorbed in the S1 segment of the PCT. This process is tightly regulated by apical transporters, most notably urate transporter 1 (URAT1) and the recently identified glucose transporter 9 (GLUT9) [116].

SGLT2 inhibition has been shown to exert renal and cardiovascular benefits not only through glucose modulation but also by lowering plasma UA concentrations [63,119]. Elevated glucose concentrations in the tubular lumen—due to SGLT2 inhibition—modulate the activity of urate-handling transporters. Of particular interest is the apical isoform of GLUT9 (encoded by the *SLC2A9b* gene), which facilitates urate/glucose exchange between the tubular lumen and epithelial cells. GLUT9-2 enables the exchange of UA for glucose in urine. With increased glucose availability in the tubule lumen, UA secretion into the tubule lumen is increased [117].

Simultaneously increased intraluminal glucose concentration suppresses UA reabsorption by transporters such as URAT1 and organic anion transporter (OAT4/OAT10). The net effect is a rise in fractional excretion of uric acid (FE-UA) and a measurable reduction in serum uric acid (PUA) [117,120].

These mechanisms’ insights are supported by robust clinical data. In a large meta-analysis involving over 31,000 participants with and without T2DM, SGLT2 inhibitors significantly reduced serum UA levels by an average of 31.5 μmol/L in diabetic individuals and by as much as 91.4 μmol/L in non-diabetic subjects [121].

The clinical implications of urate-lowering are substantial. Large-scale trials have demonstrated that SGLT2 inhibitors reduce the risk of gout flares by 30% to 50%, translating into meaningful improvements in patient quality of life, particularly among those with comorbid hyperuricemia [122].

### 5.6. Anti-Inflammatory Properties

Beyond their glycemic and uricosuric actions, SGLT2 inhibitors exhibit pronounced anti-inflammatory effects. They suppress pro-inflammatory cytokines, including interleukin-6 (IL-6), tumor necrosis factor-alpha (TNF-α), and interleukin-1β (IL-1β) [55,123,124].

Both in clinical and experimental settings, treatment with empagliflozin or canagliflozin has been demonstrated to reduce serum concentration of these cytokines [52,123,124,125].

SGLT2 inhibitors modulate intracellular signaling cascades, suppression of the nuclear factor-kappa β (NF-κB) pathway leading to decreased expression of pro-inflammatory mediators. Additionally, SGLT2 inhibition attenuates activation of the NOD-like receptor protein 3 (NLRP3) inflammasome and suppresses the inflammatory response [124,126].

Preclinical studies in rodent models have shown that dapagliflozin promotes the polarization of macrophages toward the anti-inflammatory M2 phenotype. This effect is mediated via activation of signal transducer and activator of transcription 3 (STAT3) signaling, inhibition of NF-κB, and upregulation of anti-inflammatory interleukin-10 (IL-10). This results in the reduction of cardiac fibrosis and ameliorates cardiomyocyte regeneration [127].

Treatment with SGLT2 inhibitors has an impact on adipose tissue metabolism by modulating adipokines’ synthesis. Canagliflozin has been shown to significantly increase serum adiponectin levels by 17% after 52 weeks of therapy compared to glimepiride [108]. Adiponectin enhances insulin sensitivity and exerts anti-inflammatory effects, reinforcing the metabolic benefits of SGLT2 inhibition [128,129].

Conversely, SGLT2 inhibitors reduce circulating leptin levels, with a 25% decrease observed after one year of canagliflozin treatment [128].

Given that increased leptin levels are frequently linked to obesity, chronic inflammation, and insulin resistance, this reduction may have further positive cardiometabolic effects [130].

The reduction of fat mass due to SGLT2 inhibitors results in a decrease in leptin levels and an increase in adiponectin concentrations, which also exert beneficial metabolic effects [131].

### 5.7. Combating Anemia

SGLT2 inhibitors have also demonstrated the ability to elevate hemoglobin concentration. This action begins with a decrease in plasma volume and increased hemoconcentration; however, it has been demonstrated that the improvement in anemia among patients with diabetes treated with SGLT2 inhibitors goes beyond the reduction of plasma volume [2]. Multiple hypotheses have been proposed to explain this observation. One proposed mechanism involves the promotion of erythropoiesis through hepcidin suppression, as dapagliflozin treatment reduces circulating hepcidin and ferritin concentrations while increasing levels of erythroferrone (a hepcidin inhibitor), erythropoietin (EPO), and transferrin. Collectively, these changes enhance iron availability and support red blood cell production [132].

It has been further postulated that the observed rise in hemoglobin levels may be caused by increased iron availability for hematopoiesis. Indeed, it appears that SGLT2 inhibitors promote erythropoiesis by means of improved iron management and increased EPO production, which results in a long-lasting increase in red blood cell mass [133,134].

A particularly compelling theory highlights the role of hypoxia-inducible factors (HIFs), specifically HIF-1α and HIF-2α. Due to SGLT2 inhibitors, the metabolic oxygen demand of PCT lowers, cortical renal oxygenation is improved, and local hypoxia is mitigated. This shift leads to downregulation of HIF-1α—typically upregulated under severe hypoxic or oxidative stress conditions—which results in the reduction of ROS synthesis and alleviates oxidative stress. Stable oxygen metabolism leads to preferential activation of HIF-2α. Unlike HIF-1α, HIF-2α enhances EPO production and optimizes iron metabolism, directly promoting erythropoiesis in a physiologically favorable manner [133].

A compelling confirmation of this hypothesis comes from a clinical report describing the use of tofogliflozin, an SGLT2 inhibitor, which enabled the successful discontinuation of previously administered erythropoiesis-stimulating agents (ESAs) in patients with diabetes. This observation suggests that SGLT2 inhibitors may independently stimulate erythropoiesis effectively, thereby eliminating the need for adjunctive ESA therapy. It further reinforces the role of SGLT2 inhibitors in promoting red blood cell production through mechanisms that extend beyond mere hemoconcentration [135].

### 5.8. Hepatic Effects

Emerging data suggest that SGLT2 inhibitors improve hepatic metabolism, especially in patients with T2DM with concomitant nonalcoholic fatty liver disease (NAFLD). Improvement in insulin sensitivity in this cohort was associated with decreased hepatic steatosis and the amelioration of hepatic function [18,33].

Clinical observations in a Japanese cohort showed that therapy with SGLT2 inhibitors led to a reduction in body weight and body mass index (BMI). Additionally, SGLT2 inhibitors decreased serum concentrations of HbA1c and hepatic injury markers such as alanine aminotransferase (ALT), aspartate aminotransferase (AST), and gamma-glutamyl transferase (GGTP). These benefits are further complemented by the elevation of HDL cholesterol and the decline in serum triglycerides [136].

Due to metabolic disturbances and insulin resistance in T2DM, this group is at elevated risk for developing NAFLD. Up to 17% of patients diagnosed with comorbid T2DM and NAFLD have been classified as having advanced liver fibrosis in recent epidemiological statistics, highlighting the clinical importance of therapeutic strategies targeting this axis [137]. The meta-analysis by Zhou showed that the treatment with SGLT2 inhibitors significantly reduced liver stiffness measurement (LSM) and controlled attenuation parameter (CAP), both of which are validated surrogates for hepatic fibrosis and steatosis. Importantly, these improvements were positively correlated with SGLT2 inhibitor treatment duration [19]. A reduction in liver volume was also observed in patients treated with dapagliflozin [138].

Additional imaging studies confirmed a significant decrease in hepatic fat content following SGLT2 inhibitor therapy by means of MRI-proton density fat fraction (MRI-PDFF). Although these changes did not always translate into parallel improvements in AST levels, consistent reductions in ALT were observed, indicating a favorable hepatic response [18]. The study by Dwinata confirmed these results [20].

Physiologically, SGLT2 inhibitors seem to have hepatoprotective impact, including the reduction of oxidative stress, glucagon-driven hepatic lipogenesis, systemic inflammation, and insulin resistance in addition to enhancing glycemic control. Experimental studies in animal models suggested that canagliflozin may increase the expression of hepatic zinc-α2-glycoprotein (ZAG), an adipokine involved in lipid mobilization that decreases the accumulation of hepatocellular fat [139]. Similar observations were also shown in the human population [140].

### 5.9. Potential Impact on Cognitive Function

Insulin resistance and obesity—frequent comorbidities in patients with T2DM—are increasingly recognized as contributors to cognitive dysfunction. Preclinical and clinical studies have highlighted mitochondrial impairment, enhanced oxidative stress, disrupted insulin signaling in the central nervous system, and impaired synaptic plasticity as key elements linking metabolic dysregulation to cognitive decline [141].

A 2013 meta-analysis conducted by Kapil Gudala demonstrated that individuals with diabetes had a 73% elevated risk of developing dementia compared to non-diabetics, which encompasses a 56% higher risk of Alzheimer’s Disease (AD) [142]. This is consistent with a more recent meta-analysis by Coa, which revealed that diabetic patients are 59% more likely to develop dementia than non-diabetics [143].

The neuroprotective benefits of SGLT2 inhibitors, particularly empagliflozin and dapagliflozin, were examined in a recent study by Alami using cellular models of AD. The researchers found that these inhibitors exhibited protective actions against human Aβ_1–42_-induced neurotoxicity. The study demonstrated that SGLT2 inhibitors reduced oxidative stress, decreased neuroinflammation, and attenuated tau pathology [21]. Pathological tau accumulations in neurons disrupt normal neuronal function, resulting in neuronal damage, cell death, and cognitive impairments. These processes are strongly implicated in the progression of neurodegenerative diseases [144]. These findings point out the possible therapeutic role for SGLT2 inhibitors in reducing the neurodegenerative processes linked to AD [21]. The present evidence supports the idea that SGLT2 may alleviate cognitive impairment by lowering oxidative stress and improving mitochondrial dysfunction [22,145].

Recent observational studies and meta-analyses consistently demonstrated that treatment with SGLT2 inhibitors is associated with a significant reduction in the risk of developing dementia in individuals with T2DM. A meta-analysis of five cohort studies revealed that patients receiving SGLT2 inhibitors had a significantly lower risk of dementia compared to control groups [101]. These findings suggest a potential neuroprotective effect of SGLT2 inhibitors, contributing to a decreased incidence of dementia among individuals with T2DM [23,146].

Furthermore, Youn found that SGLT2 inhibitors not only reduced the incidence of dementia but also improved cognitive outcomes, particularly in individuals with mild cognitive impairment or previously diagnosed dementia [147].

There is growing evidence that, apart from direct mechanisms of SGLT2 inhibitors, they have the molecular ability to inhibit acetylcholinesterase (AChE), thereby supporting synaptic function and neurotransmission, especially in patients with AD, as they have a reduced amount of acetylcholine neurotransmitters in the central nervous system [22].

As demonstrated in a rat model, canagliflozin reduced AChE activity and monoamine levels while improving cognitive abilities, similar to galantamine [148].

Moreover, by raising levels of brain-derived neurotrophic factor (BDNF), which is involved in the growth, survival, and plasticity of neurons and is a crucial component of learning and memorizing processes, phlozins were also found to enhance cognitive function in a diabetic mouse model [22,149,150]. Additionally, it was shown in preclinical models that SGLT2 inhibitors may also suppress seizures. They decrease neuronal glucose utilization and stabilize membrane excitability and depolarization, and thus reduce the clinical and electroencephalographic signs of epileptic seizures [151].

## 6. Adverse Effects of SGLT2 Inhibitors: A Comprehensive Clinical Overview

### 6.1. Diabetic Ketoacidosis

One of the most serious complications associated with the use of SGLT2 inhibitors is diabetic ketoacidosis (DKA). Unlike classical DKA, which is often the first sign of type 1 diabetes in adolescents and young people, DKA linked to SGLT2 inhibitors can occur while blood glucose levels are below <250 mg/dL [152]. This phenomenon is attributed to the glucosuric effect of these agents and is commonly referred to as euglycemic DKA [153].

SGLT2 inhibitors enhance α-cells to secrete more glucagon, influence pancreatic metabolism, and promote the synthesis of ketone bodies [153]. The clinical presentation of euglycemic DKA closely resembles that of classic DKA, where serum glucose levels typically remain within the normal range or are only mildly elevated [154]. The incidence of SGLT2 inhibitor-associated DKA varies across studies and agents. The DKA rate for canagliflozin and placebo in the CANVAS study was 0.6 and 0.3 events per 1000 patient-years, respectively [14]. Similarly, the EMPA-REG OUTCOME study reported a DKA incidence of <1% [9]. However, in the DECLARE-TIMI 58 trial, DKA occurred more frequently in the dapagliflozin group (0.3%) compared to placebo (0.1%) [10].

A meta-analysis by Monami reported a relative risk of 1.05 for DKA in SGLT2 inhibitor users versus controls [46]. Similarly, Wang demonstrated that elevated DKA risk was primarily observed in patients with suspected misdiagnosis of T2DM who were more likely to have latent autoimmune diabetes or type 1 diabetes. After excluding such patients, the risk disappeared, underscoring the importance of appropriate patient selection [155].

There is no clear evidence that higher doses of SGLT2 inhibitors are directly associated with an increased risk of DKA [156]. However, the canagliflozin clinical program conducted in 2015 showed a possible association. In this study, all the included patients were initially diagnosed with T2DM. The incidence of DKA was 0.522 per 1000 person-years for the 100 mg dose and 0.763 per 1000 person-years for the 300 mg dose, indicating a possible dose-related association, while the source also highlights the role of predisposing variables, such as latent autoimmune diabetes or type 1 diabetes [157].

The onset of DKA typically occurs up to 3.25 years after starting SGLT2 inhibitor therapy, with 47.6% of cases occurring during the first three months and 61.9% within the first six months. Therefore, longer duration of SGLT2 inhibitor use is not associated with an increased risk of DKA [156].

In the literature, the majority of DKA cases related to SGLT2 inhibitors occurred while using canagliflozin, which could be due to it being the first licensed SGLT2 inhibitor and having a longer duration of use in real-world clinical practice [156]. However, all SGLT2 inhibitors are thought to have a similar mechanism of action and, as a result, a comparable risk of DKA [154].

Several predisposing variables have been found, including T1DM and latent autoimmune diabetes in adults, with which T2DM can be misdiagnosed [50,156,157]. Other risk factors include acute illness (e.g., infections, gastroenteritis, heart attack, stroke), surgery, pregnancy, ketogenic diets, fasting, consuming alcohol, insulin reduction or withdrawal, and drug interactions [26,154,158,159]. In a meta-analysis by Lin, the risk of DKA was numerically more than three times higher in patients with atherosclerotic cardiovascular disease (ASCVD) than in those without, although this did not reach statistical significance. This may be due in part to a higher incidence of acute illness, such as myocardial infarction or stroke, in this subgroup [160].

There was an FDA warning about the increased risk of perioperative DKA and a recommendation to consider a withdrawal of SGLT2 inhibitors for T2DM patients three to four days prior to surgery [161]. Afterwards, it was supported by many scientific societies, among others, ACC/AHA (2024 Perioperative CV Guideline, ADA Standards of Care 2025) [162,163]. However, discontinuing SGLT2 treatments for HF patients should be conducted under strict control, as the need to keep the hold as short as safely possible and to resume early post-op was highlighted, as the beneficial influence of SGLT2 rapidly deteriorates after withdrawal in this group of patients [164].

### 6.2. Infections

The pharmacological mechanism of SGLT2 inhibitors enhances glucosuria, creating an environment conducive to bacterial colonization in the urinary and genital tracts. Although this process is widely known, data on the incidence of infections remain variable, varying depending on the site of infection [165].

#### 6.2.1. Genital Infections

A meta-analysis involving more than 50,000 participants evaluated the risk of genital infections associated with dapagliflozin, canagliflozin, and empagliflozin. The findings revealed a significantly increased risk of genital infections in the group receiving SGLT2 inhibitors compared to placebo. In women, the most frequently diagnosed disease is mycotic vulvovaginitis, and in men, mycotic balanitis [166].

These findings are further supported by additional meta-analyses. One large-scale analysis conducted by Bonnet confirmed a significantly increased risk of genital infections [131]. The consistency of this safety signal across various study populations was further supported by a pooled analysis of 3069 participants, or network meta-analysis that included 103,111 individuals [166,167].

There is evidence of a dose-response relationship for mycotic genital infections for ertugliflozin [168], although a similar dependence was not observed with other SGLT2 inhibitors [166].

Subgroup analyses in meta-analysis conducted by Liu showed that longer duration of SGLT2 inhibitor use was associated with a higher risk of genital infections [150]. On the contrary, retrospective cohort studies have shown that the risk was elevated within the first month or peaked within 90 days of initiating SGLT2 inhibitor treatment and afterwards decreased with the duration of treatment. However, long-term data on the effects of longer-term use are still being investigated [59].

Women are 4 to 5 times more susceptible to genital fungal infections than men. More events (5.2%) were reported in uncircumcised men treated with ertugliflozin compared with circumcised men (1.9%) [5,168]. A previous history of genital infections also increases the chance of developing an infection [64].

#### 6.2.2. Urinary Tract Infections

The evidence on the incidence of UTIs is less conclusive. In the meta-analysis by Liu, no statistically significant increase in UTI risk was observed between SGLT2 inhibitors and placebo. Subgroup analysis revealed a modest increase in UTI risk with dapagliflozin compared to placebo, while no significant risk of UTIs was observed for canagliflozin [32].

The quantitative meta-analysis of Lin reported a slightly but statistically significant increased risk of UTIs with SGLT2 inhibitors, consistent with earlier findings by Vasilakou [160,169].

Large cardiovascular outcome trials—EMPA-REG OUTCOME, CANVAS, and DECLARE-TIMI 58—did not report significant differences in UTIs between SGLT2 inhibitors and placebo [9,10,14].

#### 6.2.3. Fournier’s Gangrene

Fournier’s gangrene is a rare but potentially life-threatening necrotizing infection of the perineal and genital regions. Between March 2013 and January 2019, the FDA identified 55 cases of Fournier’s gangrene associated with SGLT2 inhibitors, including 3 fatalities [170].

In a retrospective cohort study of Fisher involving 208,244 participants receiving SGLT2 inhibitors and an equal number of dipeptidyl peptidase-4 inhibitor users, the incidence of Fournier’s gangrene was 0.08 per 1000 person-years in the SGLT2 inhibitors group versus 0.14 in the DPP-4 inhibitors group—a difference that did not reach statistical significance [171].

Despite its rarity, Fournier’s gangrene remains a serious adverse event requiring immediate medical attention. Clinician and patient awareness of this risk is essential for timely diagnosis and intervention.

### 6.3. Increased Risk of Fractures

Concerns were raised regarding the potential impact of SGLT2 inhibitors on bone metabolism, particularly in relation to phosphate and calcium homeostasis and the regulation of bone turnover. These theoretical mechanisms prompted investigations into whether SGLT2 inhibitors might increase the risk of impaired bone mineral density and fractures. However, data from large clinical trials and observational studies did not prove an elevated fracture risk with this drug class.

A meta-analysis of 20 randomized controlled trials (RCTs), including more than 8000 patients treated with SGLT2 inhibitors, found no statistically significant increase in fracture risk compared to placebo [172]. Similarly, another comprehensive meta-analysis encompassing 38 RCTs and over 30,000 patients reported nearly identical fracture rates in the SGLT2 inhibitor and control groups (1.59% vs. 1.56%, respectively) [173].

Additionally, in a real-world cohort study comparing users of SGLT2 inhibitors with participants receiving DPP-4 inhibitors, there was no increased risk of any fracture (adjusted HR 1.09; 95% CI: 0.91–1.31), nor of osteoporotic fractures specifically (adjusted HR 0.89; 95% CI: 0.64–1.22) [174].

Similarly, in a comprehensive network meta-analysis by Mostofa, the authors reported that SGLT2 inhibitors were not associated with an increased risk of bone fractures. On the contrary, the analysis revealed a significant reduction in fracture risk—approximately 87% in patients receiving SGLT2 inhibitors in combination with other glucose-lowering agents, and about 67% in those treated with SGLT2 inhibitors as monotherapy [175].

Among the pivotal cardiovascular outcome trials, only the CANVAS trial reported a significantly higher incidence of fractures in the group of patients treated with SGLT2 inhibitors compared to placebo [14]. In contrast, both the EMPA-REG OUTCOME and DECLARE–TIMI 58 trials reported no statistically significant differences in fracture risk [9,10].

### 6.4. Risk of Lower Limb Amputation

Among the SGLT2 inhibitors, canagliflozin was associated with an increased risk of lower limb amputation, as reported in the CANVAS trial. The incidence of amputation was 6.3 events per 1000 patient-years in the treated group, compared with 3.4 events per 1000 patient-years among the placebo patients. Most amputations involved the toe or forefoot (71%), while the minority of them involved higher-level amputations below or above the knee [14].

No significant increase in amputation risk was observed in the EMPA-REG OUTCOME or DECLARE–TIMI 58 trials, each including over 7000 and 17,000 participants, respectively [9,10]. Furthermore, data from the FDA Adverse Event Reporting System (FAERS) suggested an increased amputation risk for canagliflozin, but not for empagliflozin or dapagliflozin [176].

A meta-analysis of 14 RCTs by Li also found no overall risk among patients treated with SGLT2 inhibitors. However, subgroup analysis revealed a higher amputation risk associated specifically with canagliflozin, but not with empagliflozin [177].

Additionally, the analyses of phase I–III clinical research data on empagliflozin did not show an increased risk of amputation compared to placebo, similar to pooled data from dapagliflozin trials, which showed no increased incidence of amputation compared to placebo or other antidiabetic medications [178,179].

Moreover, empagliflozin is even recommended as a second-line agent after metformin in patients with peripheral artery disease and diabetic foot ulcers, highlighting its favorable safety profile in this population [180].

Similar conclusions were drawn in a separate large-scale meta-analysis, which noted a trend toward increased amputation risk with SGLT2 inhibitors (RR > 1) that did not reach statistical significance. Notably, the risk was considerably greater among patients with established atherosclerotic cardiovascular disease compared to those without ASCVD, which is consistent with the pathogenetic mechanism of diabetic foot development [160].

These findings are supported by a nationwide cohort study conducted by Ueda, which also found a more pronounced risk of amputation in patients with existing cardiovascular disease [176]. Importantly, pooled analyses of phase I–III clinical trial data on empagliflozin did not show an increased risk of amputation compared to placebo [178].

Evidence on ertugliflozin remains limited. However, available studies to date have not demonstrated a consistent increase in the risk of lower limb amputation with ertugliflozin [105,168].

### 6.5. Potential Association with Malignancies

SGLT2 inhibitors were investigated for their potential association with cancer, particularly bladder and breast cancer.

The CANVAS trial reported nine events of bladder cancer in the canagliflozin group against one case in the placebo group. Similarly, 12 incidences of breast cancer were recorded in the canagliflozin group compared to 8 in the group of participants receiving placebo [14].

In the EMPA-REG OUTCOME trial, five cases of bladder cancer were identified—four in the empagliflozin group and one in the placebo group [9]. In the DECLARE–TIMI 58 study, 27 cases of bladder cancer in the dapagliflozin group and 21 in the placebo group were found. Similarly, the incidence of breast cancer was comparable: 14 cases in the dapagliflozin and 13 in the placebo group [10].

A meta-analysis of 46 randomized clinical trials, encompassing 34,569 participants and identifying 580 cancer cases, found no significant increase in overall cancer risk associated with SGLT2 inhibitors. However, subgroup analysis suggested a potentially elevated risk of bladder cancer with empagliflozin, while canagliflozin was paradoxically associated with a significantly reduced risk of gastrointestinal cancer [181]. These findings are supported by a large real-world meta-analysis of randomized controlled trials involving nearly 30,000 patients treated with SGLT2 inhibitors and nearly the same number of control individuals, which found no statistically significant differences in overall cancer risk between SGLT2 inhibitor users and matched controls [182].

### 6.6. Acute Kidney Injury

In 2016, the FDA issued a safety communication warning of a potential increased risk of AKI associated with the use of SGLT2 inhibitors. This alert was based on 101 confirmed cases reported to the FAERS between 2013 and 2015, including 73 cases associated with canagliflozin and 28 with dapagliflozin, with more than half requiring hospitalization [161].

Despite these early concerns, large randomized controlled trials did not confirm an increased incidence of AKI with SGLT2 inhibitors compared to placebo. In the CANVAS, EMPA-REG OUTCOME, and DECLARE-TIMI 58 trials, the rates of AKI were similar between the treatment and placebo groups [9,10,14].

In reality, more recent and strong evidence from randomized trials and meta-analyses shows that SGLT2 inhibitors have a protective effect on renal outcomes. A 2022 meta-analysis of over 90,000 patients found that SGLT2 inhibitors significantly reduced the risk of chronic kidney disease (CKD) development and AKI compared to placebo. These advantages persisted regardless of diabetes status [183].

The results are consistent with findings from another meta-analysis showing that SGLT2 inhibitors lowered the risk of kidney disease progression by 37% and the risk of AKI by 23%, with similar benefits observed in patients with and without T2DM [13].

Furthermore, Wang confirmed that the nephroprotective effects of SGLT2 inhibitors in preventing AKI were independent of baseline renal function or concurrent use of RAAS inhibitors [184].

Although initial FAERS reports raised concerns regarding potential AKI risk, current high-quality evidence from well-conducted RCTs and meta-analyses strongly supports a renoprotective effect of SGLT2 inhibitors across various high-risk populations, including those with T2DM, HF, and CKD.

## 7. Recommendations and Guidelines from Around the World

### 7.1. Heart Failure

In the 2021 European Society of Cardiology (ESC) Heart Failure Guidelines, SGLT2 inhibitors were added to the standard treatment for patients with heart failure with HFrEF in addition to beta-blockers, mineralocorticoid receptor antagonists (MRAs), and RAAS blockers/angiotensin receptor-neprilysin inhibitors (ARNI) (Class I recommendation, Level of Evidence A), regardless of diabetic status [185].

In 2022, the American Heart Association (AHA), American College of Cardiology (ACC), and the Heart Failure Society of America (HFSA) recommended SGLT2 inhibitors as Class I (Level A) treatment for symptomatic chronic HFrEF in order to lower CV mortality and hospitalizations due to HF. Although the guidelines do not specify an eGFR cut-off beyond trial criteria, they were studied down to eGFR 20–25 mL/min/1.73 m^2^, and it is typically recommended to continue the use of SGLT2 inhibitors until advanced renal failure [186].

Evidence that led to the upgrade of SGLT2 inhibitors to a Class I, Level A recommendation also for HFmrEF was included in the 2023 ESC HF Guidelines (Focused Update). As in HFrEF, dapagliflozin and empagliflozin are recommended in HFmrEF to lower HF hospitalizations and CV mortality [187].

According to the 2022 AHA/ACC/HFSA Guidelines, SGLT2 inhibitors for HFmrEF are recommended as Class IIa, or ‘should be considered’, based on moderate quality data (Level B-R). According to the guidelines, SGLT2 inhibitors ‘may be beneficial’ in lowering HF hospitalizations and deaths in mortality due to HFmrEF [175].

Recommendations for the use of SGLT2 inhibitors in heart failure with preserved ejection fraction (HFpEF) were updated in the 2023 ESC HF Guidelines (Focused Update). Empagliflozin or dapagliflozin were the first Class I treatment for HFpEF since they showed notable risk reductions [187].

According to the 2022 AHA/ACC/HFSA Guidelines, SGLT2 inhibitors are recommended for HFpEF at Class IIa, Level B-R. In order to enhance outcomes in HFpEF, it claims that SGLT2 inhibitors ‘are reasonable to use’. Other HFpEF medications (ARNI, MRAs) were classified as Class IIb in 2022, making SGLT2 inhibitors the most strongly recommended therapy for HFpEF in those guidelines [175].

Regardless of the presence of T2DM or not, all of the HF treatment recommendations above are applicable. The studies and recommendations clearly state that SGLT2 inhibitor’s ability to lower HF events is unaffected by glucose levels.

### 7.2. Chronic Kidney Disease (CKD)

SGLT2 inhibitors are highly recommended by the KDIGO 2023 CKD Guidelines for kidney protection in individuals with CKD, regardless of T2DM. In particular, KDIGO advises using an SGLT2 inhibitor (Class I, Level A evidence) for all CKD patients with eGFR ≥ 20 mL/min/1.73 m^2^ who additionally exhibit HF or severe proteinuria (urine albumin-to-creatinine ratio ≥ 200 mg/g). Based on DAPA-CKD and EMPA-Kidney, which included non-diabetic patients, this wide 1A recommendation now covers non-diabetic CKD, building on previous recommendations for diabetic CKD. According to KDIGO, in CKD patients with or without T2D, SGLT2 inhibitor treatment significantly lowers the progression of CKD, end-stage kidney disease, and HF events.

It is advised to start with an eGFR of at least 20 mL/min/1.73 m^2^. Given the continuous renal and cardiac benefits, KDIGO recommends maintaining the treatment with SGLT2 inhibitors, even if eGFR drops below 20 mL/min/1.73 m^2^, until dialysis or intolerance. KDIGO recommends SGLT2 inhibitors for patients with lower-level albuminuria (ACR < 200 mg/g) and CKD eGFR 20–45 mL/min/1.73 m^2^, with a slightly weaker recommendation (Class IIa/B) because of limited trial evidence but possible benefit. There are no precise symptomatic criteria to start the treatment with SGLT2 inhibitors; the aim is to decrease CKD progression and lower the risk of CV/HF [188].

### 7.3. Diabetes Mellitus Type 2

Lately, there has been a shift in recommendations in the management of patients with T2DM from “glucose-centric” to “complications-centric” management. The American Association of Clinical Endocrinology (AACE) and ADA state that cardiorenal-protective therapies should be instituted independently of glycemic control. The guideline explicitly says if a patient has ASCVD, HF, or CKD, “an SGLT2 inhibitors or GLP-1 receptor analogs with demonstrated benefit is recommended as part of the regimen independently of HbA1c” [189,190].

## 8. Conclusions

SGLT2 inhibitors demonstrated remarkable therapeutic effects that extend beyond glycemic control. They provide significant benefits outside the diabetic population, altering care for millions with heart failure or CKD. Numerous mechanisms of action of SGLT2 inhibitors, such as lowering oxidative stress, inflammation, and promoting beneficial metabolic changes, provide insight into how SGLT2 inhibitors protect cells and tissue functions at the most fundamental level. The most frequent adverse events are genital mycotic infections, typically mild and responsive to standard therapy. Although euglycemic DKA is uncommon, it remains clinically significant.

With continuous study, their role in medicine is likely to expand further, providing promise for better outcomes in areas such as neurological disorders and liver disease. The robust expansion of SGLT2 inhibitors demonstrates the need for thorough outcome trials, which uncover new benefits that may improve and lengthen patients’ lives.

## Figures and Tables

**Figure 1 molecules-30-04125-f001:**
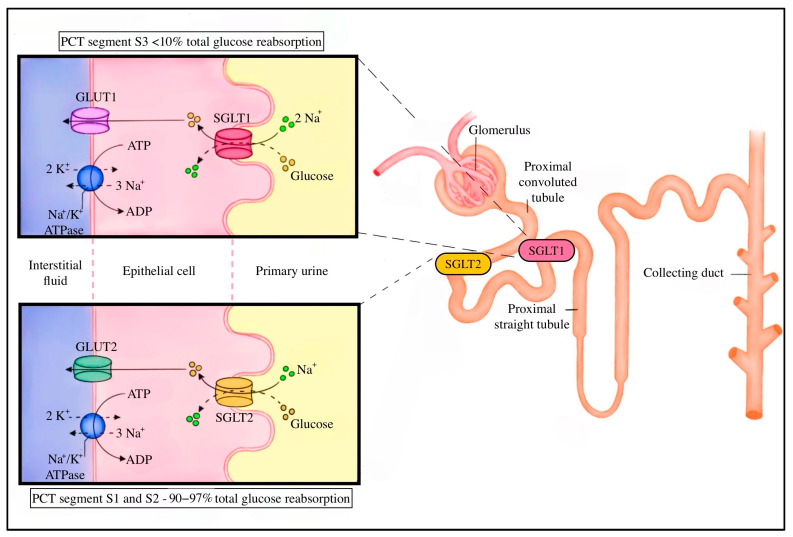
Mechanisms of renal glucose reabsorption via SGLT1 and SGLT2 in the PCT. PCT: Proximal Convoluted Tubule; SGLT1/2: Sodium-Glucose Linked Transporter 1/2; GLUT1/2: Glucose Transporter 1/2; ATP: Adenosine Triphosphate; ADP: Adenosine Diphosphate; Na^+^/K^+^-ATPase: Sodium-Potassium Adenosine Triphosphatase.

**Figure 2 molecules-30-04125-f002:**
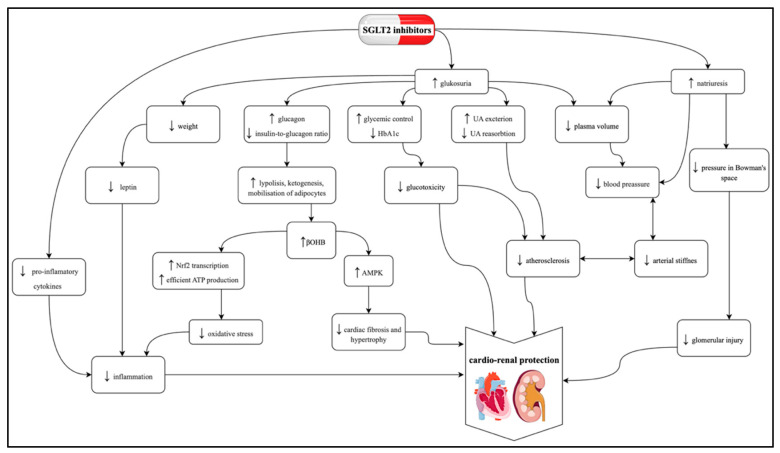
Mechanistic pathways of SGLT2 inhibitors leading to cardio-renal protection. AMPK: Adenosine Monophosphate-Activated Protein Kinase; ATP: Adenosine Triphosphate; βOHB: beta-Hydroxybutyrate; HbA1c: Hemoglobin A1c; Nrf2: Nuclear Factor Erythroid 2–Related Factor 2; UA: Uric Acid. Arrows within boxes (↑ or ↓) signify the direction of alteration of a specified parameter: ↑—increase, activation, or stimulation of a process; ↓—decrease, inhibition, or suppression of a process. Arrows between boxes (→) denote causal linkages or downstream consequences, and bidirectional arrows (↔) signify mutual or feedback interactions. Image adapted from Servier Medical Art (https://smart.servier.com), licensed under CC BY 4.0 (https://creativecommons.org/licenses/by/4.0/ URL accessed on 23 July 2025).

**Table 1 molecules-30-04125-t001:** Key clinical trials of SGLT2 inhibitors (and SGLT1/2): cardiovascular/renal efficacy and safety.

Study	Year	Drug	Condition/ Population	Median Follow-Up	Primary Endpoint	Overall Result	Key Adverse Events/ Complications (Reported)
EMPA-REG OUTCOME [9]	2015	Empagliflozin 10 mg/25 mg	T2DM with ASCVD (7020 patients)	3.1 years	3-point MACE	↓ MACE (HR ≈ 0.86); large ↓ CV death (~38%); ↓ HHF	↑ genital mycotic infections
CANVAS Program [14]	2017	Canagliflozin 100 mg/300 mg	T2DM with ASCVD or high CV risk (10,142 participants)	3.6 years (mean 188 weeks)	3-point MACE	↓ MACE (HR ≈ 0.86); renal-benefit signals	↑ lower-limb amputations (HR~1.97) and fracture signal; ↑ genital mycotic infections
DECLARE–TIMI 58 [10]	2019	Dapagliflozin 10 mg	T2DM with multiple risk factors or ASCVD (17,160 patients)	4.2 years	Dual: MACE; CV death or HHF	Neutral MACE; ↓ CV death/HHF (HR~0.83)	↑ DKA (0.3% vs. 0.1%); ↑ serious genital infections
VERTIS CV [105]	2020	Ertugliflozin 5 mg/15 mg	T2DM with established ASCVD (8246 patients)	3.5 years (mean)	3-point MACE (non-inferiority)	Non-inferior for MACE; trend ↓ HHF	↑ genital mycotic infections/UTIs vs. placebo; overall safety otherwise balanced
DAPA-HF [106]	2019	Dapagliflozin 10 mg	HFrEF (with/without diabetes) (4744 patients)	18.2 months	CV death or worsening HF	↓ primary composite (HR~0.74)	↑ genital infections; DKA rare
EMPEROR-Reduced [8]	2020	Empagliflozin 10 mg	HFrEF (with/without diabetes) (3730 patients)	16 months	CV death or HHF	↓ primary composite (HR~0.75)	↑ hypotension/volume depletion and genital infections
EMPEROR-Preserved [107]	2021	Empagliflozin 10 mg	HFpEF/HFmrEF (LVEF > 40%) 5988 patients	26.2 months	CV death or HHF	↓ primary composite (HR~0.79)	↑ genital/urinary infections and hypotension vs. placebo
DELIVER [108]	2022	Dapagliflozin 10 mg	HFpEF/HFmrEF (LVEF > 40%) 6263 patients	2.3 years	Worsening HF or CV death	↓ primary composite (HR~0.82)	↑ genital infections; hypotension
DAPA-CKD [109]	2020	Dapagliflozin 10 mg	CKD with/without T2DM 4304 participants	2.4 years	≥50% eGFR decline, ESKD or kidney/CV death	↓ renal composite and ↓ all-cause mortality	↑ genital mycotic infections; DKA rare
EMPA-KIDNEY [110]	2023	Empagliflozin 10 mg	CKD with/without diabetes (broad eGFR/albuminuria) 6609 patients	2.0 years	Kidney disease progression or CV death	↓ primary composite (HR~0.72)	AKI events not increased; ↑ genital infections (class-typical)
CREDENCE [12]	2019	Canagliflozin 100 mg	T2DM with CKD (albuminuric) 4401 patients	2.62 years	ESKD, doubling of creatinine, or renal/CV death	↓ primary renal composite (HR~0.70)	No significant ↑ amputations or fractures vs. placebo; ↑ male genital infections; rare DKA
EMPULSE [111]	2022	Empagliflozin 10 mg	Acute HF (initiated in-hospital, stabilized 566 patients	90 days (hierarchical endpoint)	Win ratio composite (death, HF events, time to first HF event)	Significant clinical benefit at 90 days (win ratio ~1.36)	Well tolerated
EMPACT-MI [112]	2024	Empagliflozin 10 mg	Recent MI with high HF risk (without established HF) 3260 patients	17.9 months	First HHF or all-cause death	Neutral for primary endpoint; HHF component reduced	AEs similar between groups; class-consistent safety
DAPA-MI [113]	2024	Dapagliflozin 10 mg	Recent MI without diabetes or chronic HF 4017 patients	≈1 year	Hierarchical cardiometabolic win composite	Win for cardiometabolic outcomes; no difference in CV death/HHF	AEs comparable to placebo; genital infections uncommon in non-diabetics
SOLOIST-WHF (SGLT1/2) [114]	2021	Sotagliflozin 200 mg/400 mg	T2DM with recent worsening HF 1222 patients	≈9 months	Total CV death + HF hospitalizations/urgent visits	↓ total CV death/HF events	Class-typical genital infections
SCORED (SGLT1/2) [115]	2021	Sotagliflozin 200 mg/400 mg	T2DM with CKD and CV risk 10,584 patients	≈16 months	CV death + HF hospitalization/urgent visits	↓ primary composite; cardiorenal benefits in analyses	↑ diarrhea, genital mycotic infections, volume depletion, and DKA vs. placebo

AE/AEs—adverse event (s); AKI—acute kidney injury; ASCVD—atherosclerotic cardiovascular disease; CKD—chronic kidney disease; CV—cardiovascular; DKA—diabetic ketoacidosis; eGFR—estimated glomerular filtration rate; ESKD—end-stage kidney disease; HF—heart failure; HFmrEF—heart failure with mildly reduced ejection fraction; HFpEF—heart failure with preserved ejection fraction; HFrEF—heart failure with reduced ejection fraction; HHF—hospitalization for heart failure; HR—hazard ratio; LVEF—left ventricular ejection fraction; MACE (3-point MACE)—major adverse cardiovascular events; here: CV death, nonfatal MI, nonfatal stroke; MI—myocardial infarction; T2DM—type 2 diabetes mellitus; UTI—urinary tract infection. An upward arrow (↑) indicates an increase in the incidence of events, while a downward arrow (↓) indicates a decrease in the specified outcome or event.

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
