# Peer review of "Sodium-Glucose Cotransporter-2 Inhibitors in Diabetes and Beyond: Mechanisms, Pleiotropic Benefits, and Clinical Use—Reviewing Protective Effects Exceeding Glycemic Control"

_molecules, 2025, doi:10.3390/molecules30204125_

Round 1

Reviewer 1 Report

Comments and Suggestions for Authors
  1. The first subtitle in the Introduction is missing.
  2. In the “Historical Background and Development of SGLT2 Inhibitors” section, the relationship between phlorizin and SGLT2 should be clearly explained.
  3. Please explain the distinct roles and distributions of SGLT2 and SGLT1 in glucose reabsorption.
  4. In the “Mechanism of Action and Pharmacodynamics of SGLT2 Inhibitors” section, it is recommended to use subheadings to clarify the various mechanisms.
  5. In the “Genomic Localization and Structure of SGLT Family Members” section, the relevance of this paragraph to the focus on SGLT2 should be clarified. Additionally, since SGLT3 is described as a glucose sensor rather than a transporter, is this functional difference related to specific variations in its gene structure, such as in the promoter or coding regions?
  6. The quality of Figure 2 needs improvement.
  7. Overall, the authors attempted to cover extensive aspects of SGLT2 inhibitors but lacked in-depth discussion and a clear focus, which reduced the readability of the work. It would be better to focus on one or two themes and provide more thorough analysis and perspective.
  8. According to the latest guidelines, in which non-diabetic populations are SGLT2 inhibitors recommended, and what are the recommendation and evidence levels?
  9. Section five discusses the effects on blood glucose, weight, cardiovascular and renal outcomes, uric acid, inflammation, anemia, liver, and cognitive function. These should be more closely connected to the preceding topics.

Author Response

Thank you very much for your careful reading our manuscript and for your comments.

Comment 1: The first subtitle in the Introduction is missing.
Response 1:The first subtitle in the Introduction is now in place; its absence was due to a formatting oversight at submission.

Comment 2.    In the “Historical Background and Development of SGLT2 Inhibitors” section, the relationship between phlorizin and SGLT2 should be clearly explained.
Response 2: The relationship was explained. 

Comment 3.    Please explain the distinct roles and distributions of SGLT2 and SGLT1 in glucose reabsorption.
Response 3: We have explained the roles and distributions of SGLT2 and SGLT1 in glucose reabsorption: lines 96-107. 

Comment 4:    In the “Mechanism of Action and Pharmacodynamics of SGLT2 Inhibitors” section, it is recommended to use subheadings to clarify the various mechanisms.
Response 4: We have added subheadings throughout the ‘Mechanism of Action and Pharmacodynamics of SGLT2 Inhibitors’ section according to the Reviewer’s suggestion.

Comment 5. In the “Genomic Localization and Structure of SGLT Family Members” section, the relevance of this paragraph to the focus on SGLT2 should be clarified. Additionally, since SGLT3 is described as a glucose sensor rather than a transporter, is this functional difference related to specific variations in its gene structure, such as in the promoter or coding regions?

Response 5: We have refocused the “Genomic Localization and Structure” section to be explicitly SGLT2-centric. We also deleted the SGLT3 subsection, as it did not materially advance the SGLT2-focused narrative.

Comment 6.    The quality of Figure 2 needs improvement.
Response 6: The quality of Figure 2 was improved.

Comment 7.    Overall, the authors attempted to cover extensive aspects of SGLT2 inhibitors but lacked in-depth discussion and a clear focus, which reduced the readability of the work. It would be better to focus on one or two themes and provide more thorough analysis and perspective.
Response 7: While our initial scope centered on renal protection and cardiovascular benefit, the literature review revealed other clinically relevant advantages of SGLT2 inhibitors, and we thus decided to present their multivarious functions.

Comment 8.    According to the latest guidelines, in which non-diabetic populations are SGLT2 inhibitors recommended, and what are the recommendation and evidence levels?
Response 8: We have added a paragraph which now compiles worldwide recommendations for SGLT2 inhibitors use in non-diabetic populations, listing the recommendation classes and evidence levels for heart failure across the ejection fraction spectrum and for CKD (see now Section 7). For completeness, we also note diabetes-focused guidance separately.

Comment 9.    Section five discusses the effects on blood glucose, weight, cardiovascular and renal outcomes, uric acid, inflammation, anemia, liver, and cognitive function. These should be more closely connected to the preceding topics.
Response 9: Section five decribes clinical functions of SGLT2 inhibitors and thus we wanted to emphasise the numerous roles of SGLT2 inhibitors.

Reviewer 2 Report

Comments and Suggestions for Authors

The article was meant to  review/update the known remarkable and diverse beneficial effects provided by the clinical use of the SGLT2 inhibitors, which are relatively novel and have been approved for antidiabetic therapy for several years.  However, potential adverse effects were noted. This article also summarized the side effects revealed by a few reports.   Since this is a review article, none of the requirements regarding Methods, Results are relevant here. Granted, the beneficial activities for individual organs or systems such as for the heart, the kidney and liver, and for the neural and circulating systems, have been reviewed separately by several articles from the last 3-5 years.  Likewise certain prominent side effects were separately published. My assessments for the article are the followings:

1.    The article is meant to provide a thorough overview or update of the mechanisms and diverse  beneficial effects,  as well as the adverse effects  revealed by the clinical uses of the SGLT2 inhibitors.  For this purpose, the authors have done a commendable effort to summarize all the available information derived from both the  basic and  clinical studies.

2.    Compared to all the other published reports, the article provided a much more comprehensive and thorough presentation.

3.    Specific improvements required by this reviewer: It is deemed important  if certain prominent  adverse effects such as diabetic ketoacidosis, increased risks of  bone  fracture and malignancy might be associated with higher  doses  and/or longer duration of use of the SCLT2 inhibitors; and if specific subgroup of the agents might be more prevalent.

4.    In general, the Conclusion is consistent with the evidence and arguments presented, although additional discussion may be expanded to address the potential negative impacts and safety issues due to the side effects.

5.    The references are appropriate. The figures presented seem to be OK.

Author Response

Thank you very much for your careful reading our manuscript and for your comments.

Comment 1:  The article is meant to provide a thorough overview or update of the mechanisms and diverse  beneficial effects,  as well as the adverse effects  revealed by the clinical uses of the SGLT2 inhibitors.  For this purpose, the authors have done a commendable effort to summarize all the available information derived from both the  basic and  clinical studies.
Response 1: Thank you very much for your positive comment.

Comment 2: Compared to all the other published reports, the article provided a much more comprehensive and thorough presentation.
Response 2: We thank the Reviewer for this generous comparison. 

Comment 3:    Specific improvements required by this reviewer: It is deemed important  if certain prominent  adverse effects such as diabetic ketoacidosis, increased risks of  bone  fracture and malignancy might be associated with higher  doses  and/or longer duration of use of the SGLT2 inhibitors; and if specific subgroup of the agents might be more prevalent.
Response 3: Following the reviewer’s recommendation, we expanded the section on adverse effects of SGLT2 inhibitors to include (i) potential and confirmed dose–response and treatment-duration associations, (ii) identification of patient subgroups most susceptible to these events, and (iii) additional context that deepens the discussion.

Comment 4:    In general, the Conclusion is consistent with the evidence and arguments presented, although additional discussion may be expanded to address the potential negative impacts and safety issues due to the side effects.
Response 4: We revised the Conclusion section to include a concise safety statement that balances efficacy with key risks and mitigation.

Comment 5:    The references are appropriate. The figures presented seem to be OK.
Response 5: Thank you very much for your comment.

Reviewer 3 Report

Comments and Suggestions for Authors

Reviewer comments and suggestions

The authors in this study discussed the SGLT2 inhibitors, beyond their glucose-lowering action, offer significant cardiovascular, renal, and metabolic benefits. Despite some safety concerns, their favorable risk–benefit profile supports their expanding role in clinical practice. Overall, they represent a paradigm shift from conventional glycemic control to holistic organ protection in diabetes and related disorders.

Decision: Major revision

I recommend incorporating the following points into the revised version of the manuscript:

  1. Line 33-39 the first 4-5 lines cited 13 references that seemed to be odd. It would be nice if the authors discussed few relevant references and then cite them at their place
  2. Line 60 a typo error was seen, please update accordingly
  3. Figure 2 needs to be clearer and well established phenomena can be used
  4. Line 223-224 is better to discuss and explain about the cited references
  5. Line 304-305 mechanism should be well described
  6. Line 692-698 the lines should be justified and in a single paragraph
  7. Please prepare two tables that can relate with using SGLT2 inhibitors in type 2 diabetes and CVD with the help of published clinical trial
  8. Almost all the references need to be modified, please check the MDPI journal guidelines

Author Response

Thank you very much for your careful reading our manuscript and for your comments. 

Comment 1: Line 33-39 the first 4-5 lines cited 13 references that seemed to be odd. It would be nice if the authors discussed few relevant references and then cite them at their place
Response 1: Thank you for noting this. Those sentences were intended as a high-level signpost of the manuscript’s scope, hence the broad set of citations. We selected to keep lines 33–39 unchanged, as the Introduction serves as a concise roadmap supported by high-level references. Detailed discussion and placement of the individual citations occur in Section 5, which substantially expands on these points.

Comment 2: Line 60 a typo error was seen, please update accordingly.
Response 2: The mistake was corrected.

Comment 3: Figure 2 needs to be clearer, and well-established phenomena can be used.
Response 3: The quality of Figure 2 was improved.

Comment 4: Line 223-224 is better to discuss and explain about the cited references.
Response 4: Thank you for your valuable comment. We revised the section to provide a more detailed discussion of the cited studies. In the revised version, we describe the design and main findings of the key clinical trials (Refs. 55 and 56), emphasizing their contribution to the evidence supporting the efficacy of SGLT2 inhibitors in combination with insulin therapy.

Comment 5: Line 304-305 mechanism should be well described
Response 5: Thank you for the suggestion. We have expanded and clarified the mechanism in this section explaining that SGLT2i and ACEi both lower intraglomerular pressure (afferent vs. efferent), accounting for the early eGFR dip, slower long-term decline, reduced proteinuria, and podocyte protection; dapagliflozin preclinical data are cited

Comment 6: Line 692-698 the lines should be justified and in a single paragraph
Response 6: We have corrected the formatting—lines 692–698 now appear as one justified paragraph.

Comment 7; Please prepare two tables that can relate with using SGLT2 inhibitors in type 2 diabetes and CVD with the help of published clinical trial.

Response 7: We have added a table summarizing key published study results and reported side effects from that study.

Comment 8: Almost all the references need to be modified, please check the MDPI journal guidelines
Response 8: We have corrected the references according to MDPI requirements.